# Measuring Frequency Non-Stationarity for Robust Time Series Forecasting

## Abstract

Non-stationarity in time series has long posed a fundamental challenge for forecasting models, as it leads to distribution shifts between training and test data. A popular line of research, known as normalization methods, aims to measure and suppress non-stationarity by removing time-domain low-order statistics. Nevertheless, low-order statistics may inadequately address the underlying non-stationary structures manifested as a composition of frequencies. To tackle these issues, we propose to measure the degree of stationarity of each frequency component across distributions via spectral analysis. By identifying and downweighting frequencies that are more non-stationary, we re-represent the original time series to reduce distributional discrepancies between training and test sets. Concretely, we present FREMEN with threefold contributions. *Theoretically*, FREMEN is grounded in a principled formulation and we provide the first spectral analysis to support its validity. *Technically*, FREMEN is both novel and effective, incurring negligible additional computational cost. *Experimentally*, FREMEN is validated on four forecasting models across seven datasets, achieving 24 best results out of 28 settings and 28.4% average MSE improvements. Our code is publicly available[1].

## 1 Introduction

Time series forecasting is vital to decision-making in real-world applications like industrial system control and stock market tracking (Thompson & Wilson, 2016; Zhao et al., 2024). Recently, deep learning has shown some promise on benchmark datasets (Nie et al., 2023; Liu et al., 2024; Piao et al., 2024b; Wang et al., 2025). However, a challenge remains: *the non-stationary nature of time series such as seasonal fluctuations and irregular events often leads to poor generalization when forecasting models are applied to unseen test data.* The non-stationarity baffles training-patterns-driven forecasting models that assume consistency in the test dataset. Therefore, when the distribution shift occurs, these models show forecasting degeneration.

To tackle this issue, a recent popular line of research focuses on normalization methods that aim to *measure* and suppress non-stationarity in the input samples, thereby reducing distributional discrepancies between the training and test datasets (Kim et al., 2021; Liu et al., 2023; Fan et al., 2023; Han et al., 2024; Ye et al., 2024). These methods explicitly measure non-stationarity through statistical metrics (typically mean and variance) computed or learned from the training set. The metrics are then used to normalize the input, attempting to remove distribution shifts manifested in the location and scale. Since the normalization is applied consistently during both training and inference, it helps align the data distributions and thus improves generalization. Importantly, many of these methods learn to control the normalization strength through adaptive gates (Fan et al., 2023) or residual connections (Liu et al., 2023), balancing between preserving the original distributional information and measuring statistical stability for more robust forecasting.

Nevertheless, existing methods estimate statistics in the time domain, which may inadequately measure the non-stationarity due to the following reasons. First, these methods primarily rely on normalizing raw temporal values. While it may be effective for simple scale and temporal variations, it often fails to account for more complex *non-stationary structures* in the frequency domain, such as temporal drift in dominant frequencies, spectral reallocation, or shifts in periodicity (Ye et al., 2024;

---

[1] https://anonymous.4open.science/r/Fremen-code-82C8

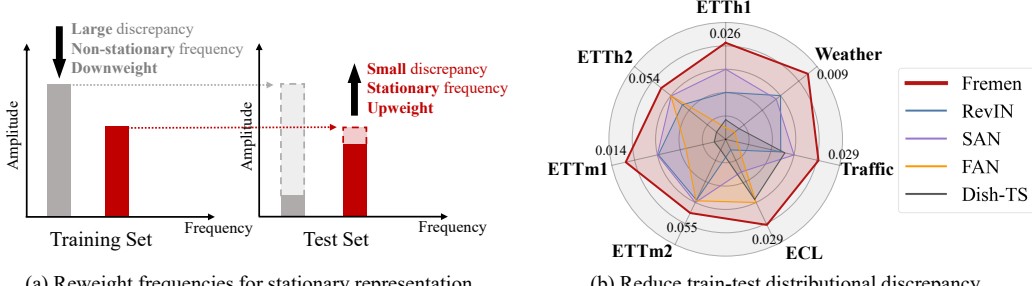

Figure 1: (a) The objective of this paper is to measure stationarity of frequencies and reweight them accordingly for stationary representation. (b) The learnt representation achieves the lowest Jensen-Shannon divergence between the training and test samples compared to existing methods. A larger shaded area indicates a smaller distributional discrepancy between the training and test set.

Piao et al., 2024b). As a result, these methods struggle to mitigate distribution shifts that consist of the underlying spectra of time series data. Second, the use of low-order statistics is insufficient to characterize complex distributional structures and intricate temporal dependencies (e.g., multi-modality, higher-order statistics, or changes in its functional form) (Han et al., 2024). Consequently, the normalization becomes inaccurate, hindering the effectiveness in measuring distribution shifts.

To address these issues, this paper presents FREMEN, a frequency-space, non-stationarity-aware method to mitigate distribution shifts in time series forecasting. As shown in Figure 1, our idea is to perform spectral analysis to measure the degree of stationarity of each frequency component across samples. By identifying and downweighting frequency components that are more non-stationary, we re-represent the original time series to reduce distributional discrepancies between training and test sets. While a recent study also attempts to analyze non-stationarity in the frequency domain (Ye et al., 2024). It heuristically selects the top-$k$ frequency magnitudes, running the risk of low-frequency dominance and inadequately characterizing the entire spectrum. By contrast, we introduce a kernel representation that is implicitly induced by the Fourier transform, integrating out all possible distribution shift patterns via the Yaglom's theorem (Yaglom, 1987). By the one-one correspondence between kernel and spectral weights, learning the weights of frequencies is equivalent to learning a data-adaptive kernel representation itself, allowing the model to capture subtle distributional discrepancies and prioritize stationary frequency components for improved generalization.

Our contributions are threefold. Theoretically, we provide the first spectral analysis of frequency-domain non-stationarity to combat distribution shift. Methodologically, we present a simple yet effective algorithm for learning to weight the frequency components, thereby re-representing the distribution behind time series data. Experimentally, FREMEN is validated on four mainstream forecasting backbone models across seven benchmark datasets, achieving 24 best results out of 28 settings and 28.4% average MSE improvements in multivariate forecasting.

## 2 RELATED WORKS

**Non-stationary Time Series Modeling.** Existing methods mainly aim to find a way to measure the distribution shift in the time domain, thereby helping the model learn a robust data representation (Du et al., 2021; Kim et al., 2021; Piao et al., 2024a). RevIN (Kim et al., 2021) proposed using mean and variance to measure the distribution shift. They first set the mean and variance of each sample to a fixed value. After forecasting, the original values are returned to the forecasting model outputs. A series of time-domain methods followed this idea: (i) Earlier works consider the evolution of mean and variance between inputs and outputs and explicitly model them (Fan et al., 2023; Liu et al., 2023). (ii) Recent works tend to learn a more expressive measure of non-stationarity than low-order statistics (Han et al., 2024; Liu et al., 2023). Recently, FAN (Ye et al., 2024) took an initial step toward addressing distribution shifts in the frequency domain. FAN identifies high-amplitude frequencies as unstable and seeks to mitigate distribution shifts by removing these frequencies. However, this heuristic strategy may risk discarding critical patterns and lead to suboptimal performance.

**Frequency Domain Modeling.** Non-stationary time series can be seen as a mix of frequencies that vary over time (Proakis & Manolakis, 1996). Earlier methods often aim to learn frequency features directly from the raw Fourier coefficients (Wu et al., 2021; Zhou et al., 2022b; Wang et al., 2022; Wu et al., 2023; Yi et al., 2023). However, the frequency features are often sparse and mixed with noise and time-varying features (Proakis & Manolakis, 1996; Piao et al., 2024b). Recent methods tend to learn more informative and robust representations via sparse selection (Zhou et al., 2022b; Woo et al., 2022; Zhou et al., 2022a; Ye et al., 2024) or normalization (Piao et al., 2024b). However, these methods often rely on heuristic strategies and do not consider cross-sample variations. To the best of our knowledge, we are the first to model the frequency variations across samples to mitigate the impact of non-stationary features on forecasting.

# 3 PROBLEM SETTING AND PRELIMINARY ANALYSIS

We formulate the problem of distribution shifts in non-stationary time series in Section 3.1, followed by our novel theoretical analysis in Section 3.2. Based on the analysis, we present a novel forecasting method FREMEN in Section 4. The key notations used in the paper are summarized in Table 1.

## 3.1 PROBLEM SETUP

**Definition 1 (Time series data and forecasting.)** We consider the multivariate time series forecasting problem on a given dataset $\{\mathcal{X}, \mathcal{Y}\}$, with $\mathcal{X} = \{x^{(i)}\}_{i=1}^N, \mathcal{Y} = \{y^{(i)}\}_{i=1}^N$ and $N$ denotes the number of sequences. Let $C, L_x, L_y$ respectively denote the number of variables, the input-sequence length and the model prediction length, then the goal can be formulated as that given an input time series data $x^{(i)} \in \mathbb{R}^{L_x \times C}$, predict the target values $y^{(i)} \in \mathbb{R}^{L_y \times C}$.

**Definition 2 (Distribution shift issue in forecasting.)** We consider the forecasting under distribution shift issue induced by non-stationarity in time series data. We assume that the data $\mathcal{X}$ is generated from an evolving distribution over time $P_t(x)$. A time series is said to be *stationary* if its distribution remains invariant over time, i.e., $P_{t_1}(x) = P_{t_2}(x)$ for all $t_1, t_2$. Conversely, *non-stationarity* refers to scenarios where the distribution changes with time: $\exists t_1 \neq t_2$ s.t. $P_{t_1}(x) \neq P_{t_2}(x)$. Such distribution shifts can manifest through changes in the mean, variance, feature correlation, or other latent structure of the input sequences. Formally, given a training set $\mathcal{X}_{\text{train}} = \{x^{(i)}\}_{i=1}^{N_{\text{train}}}$ drawn from $P_t(x)$ with $t \in \mathbb{T}_{\text{train}}$, the goal is to make accurate predictions on future inputs $x$ drawn from a different distribution $P_{t'}(x)$ with $t' \in \mathbb{T}_{\text{test}}$, $\mathbb{T}_{\text{test}} \cap \mathbb{T}_{\text{train}} = \emptyset$, and $P_{t'}(x) \neq P_t(x)$.

Table 1: Key notations used in this paper.

| Notation | Description |
|---|---|
| $\mathcal{X}, \mathcal{Y}$ | input and target time series |
| $L_x, L_y$ | input sequence length, prediction length |
| $N$ | number of variables |
| $C$ | number of channels |
| $P_t$ | time series generation distribution |
| $t$ | sample index of time series |
| $\omega$ | frequency component |
| $\mathcal{S}$ | power spectral density |
| $k$ | valid kernel |
| $\lambda$ | eigenvalue, frequency weight |
| $F$ | frequency coefficient |

**Problem (Statistical non-stationarity measure.)** A common paradigm is the use of statistical normalization techniques applied directly to the input sequences. These methods normalize the observations across the temporal dimension before feeding into the model. Formally, it computes channel-wise $\mu_t, \sigma_t$ (e.g., mean and standard deviation at time $t$), and transforms $x$ into a normalized time series $\tilde{x}$: $\tilde{x}_{t,c} = \frac{x_{t,c} - \mu_{t,c}}{\sigma_{t,c}}, \quad \forall t \in [1, L_x], c \in [1, C]$. $\mu_t, \sigma_t$ can be empirically computed (Kim et al., 2021), learned (Fan et al., 2023), or vectorized using sliding windows (Liu et al., 2023).

However, this paradigm implicitly assumes that the data generating distribution $P_t(x)$ can be fully characterized by its low-order statistics. This assumption rules out the possibility of more complex distributions. Even in the location-scale family, members like the Student's $t$ distribution depend on additional parameters (Zhu et al., 2025). Existing normalization methods thus fail to reflect complex non-stationary patterns like frequency shift, temporal dynamics, or latent structural changes.

Therefore, *our goal is to develop more expressive, learnable measures of non-stationarity* that can adaptively characterize evolving dynamics in the input time series.

## 3.2 PRELIMINARY ANALYSIS: MEASURING NON-STATIONARITY IN FREQUENCIES

We investigate non-stationarity in time series from a frequency-domain perspective. The Fourier transform decomposes a time series into basis functions, disentangling temporal structure into interpretable frequency bands. Our theoretical analysis first shows that variations in the power spectrum reflect underlying distribution shifts (Lemma 1), and that such spectral differences provide a valid non-stationarity measure (Lemma 2). We further prove that modeling these differences induces a shift-invariant kernel in the frequency domain, offering a principled way to emphasize stationary components for robust forecasting (Lemma 3). We begin by formally stating our main theorem.

**Theorem 1** *The Fourier transform on the timeseries dataset $\mathcal{X}$ induces a similarity measure $k$ that is invariant to the non-stationarity. This measure $k$ can be learned in a data-driven manner by learning its frequency weights $\{\lambda_i\}$.*

To support this theorem, we present the following lemmas, which respectively establishes that: (i) spectral representations encode key differences to non-stationarity not apparent in the time domain; (ii) a measure capable of gauging the differences in the spectral domain holds the potential of distinguishing distribution shifts; (iii) kernel function is a valid measure that can be adapted to data by identifying its eigenvalues.

**Lemma 1 (Spectral shift and energy redistribution)** *Let $x_t$ and $x'_t$ be two sampled non-stationary time series, where the distribution shifts over time. Then their spectral representations $\hat{x}$ and $\hat{x}'$ exhibit distinct energy distributions across frequency bands:*

$$\exists \omega \quad s.t. \quad |\hat{x}(\omega)|^2 \neq |\hat{x}'(\omega)|^2.$$

*This redistribution of spectral energy reflects the underlying non-stationary behavior (e.g., seasonal transitions, structural drifts), which may not be apparent in the time domain.*

Thus, power spectral density (PSD) analysis may provide a principled way to quantify time-varying distributions. We next formalize the discriminative capability of these spectral patterns, thereby validating the use of frequency-domain representations as a principled measure of non-stationarity.

**Lemma 2 (Discriminative power of spectral distribution)** *Let $\mathcal{X}_1, \mathcal{X}_2$ be two subsets of sequences drawn from distributions $P_{t_1}$ and $P_{t_2}$ respectively, with $P_{t_1} \neq P_{t_2}$. Assume their average power spectral densities are $\mathcal{S}_1(\omega)$ and $\mathcal{S}_2(\omega)$. Then the total variation distance between them satisfies:*

$$\text{TV}(\mathcal{S}_1, \mathcal{S}_2) = \frac{1}{2} \int |\mathcal{S}_1(\omega) - \mathcal{S}_2(\omega)| \, d\omega > 0.$$

*This implies that frequency-domain statistics can effectively distinguish different time-evolving distributions, and thus serve as a valid non-stationarity measure.*

Having established that spectral differences can measure non-stationarity, we next explore how to learn a function to model the difference in a principled way. We turn to spectral analysis to show that a valid similarity measure as a kernel function is implicitly induced by Fourier transform.

**Lemma 3 (Yaglom's Theorem)** *A continuous bounded function $k$ on $\mathbb{R}^{L_x}$ is a valid kernel if and only if it can be represented as*

$$k(x_1, x_2) = \int_{\mathbb{R}^{L_x} \times \mathbb{R}^{L_x}} e^{2\pi i (\omega_1^\top x_1 - \omega_2^\top x_2)} \mathcal{S}(\omega_1, \omega_2) \mathrm{d}\omega_1 \mathrm{d}\omega_2$$

*where $\mathcal{S}(\omega_1, \omega_2)$ can be understood as a joint probability density function (Yaglom, 1987). Because a kernel can be fully characterized by its eigen-decomposition (Scholkopf & Smola, 2001), Yaglom's theorem indicates that the measure $k$ induced by Fourier transform on data can be adapted to data by learning its eigenvalues $\{\lambda_i\}$.*

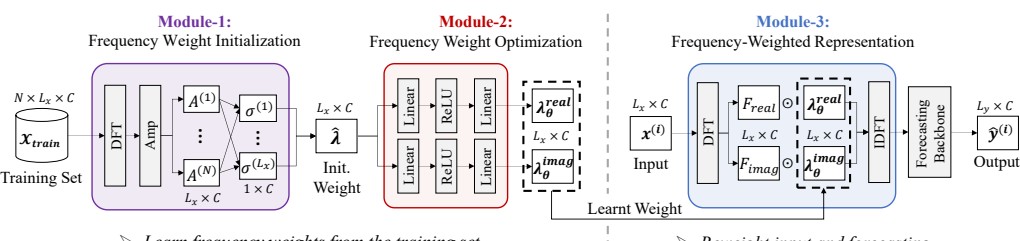

Figure 2: FREMEN begins by initializing frequency weights from the training set and refining them through neural network optimization (*Module-1,2*). Then, given an input, the weights are applied to produce a frequency-weighted input for the forecasting backbone to perform prediction (*Module-3*).

As time-sequential data can be derived from the integration of harmonic waves, the Yaglom's theorem implies that a kernel as the result of integrating over the distribution of power spectra, is invariant to the time-varying statistical characteristics of time series (Xue et al., 2023).

In summary, our theorem and the lemmas manifest a key point: when we move to the frequency domain, the differences in power spectra directly capture non-stationarity. This means we can think of the kernel as a mathematical tool that measures similarity between time series in the frequency domain. Frequency components capture patterns at different scales, and kernels provide a principled way to weight them by their stability across training and test data. This naturally emphasizes stationary frequencies while suppressing non-stationary ones. Our goal is to identify relatively stationary frequency weights to mitigate distribution shift in non-stationary time series forecasting.

## 4 PROPOSED METHOD: FREMEN

Based on our analysis in Section 3.2, we present a novel forecasting framework FREMEN in this section. FREMEN can be employed as a representation layer that reweights frequency components of the input to produce non-stationarity-aware features.

### 4.1 OVERALL FORWARD PROCESS

The forward process is summarized in Figure 2. FREMEN first learns frequency weights from the training set $\mathcal{X}_{\text{train}}$. Specifically, the *frequency weight initialization (Module-1)* takes samples in $\mathcal{X}_{\text{train}}$ as input, outputs the empirical kernel eigenvalue $\hat{\lambda}$ as initial frequency weights. Then, $\hat{\lambda}$ is fed to the neural network in the *frequency weight optimization (Module-2)*, producing weights for real and imaginary parts, i.e., $\lambda_\theta^{\text{real}}$ and $\lambda_\theta^{\text{imag}}$. Finally, when a new input $x$ arrives, the *frequency-weighted representation (Module-3)* transforms it into the frequency domain, applies $\lambda_\theta^{\text{real}}$ and $\lambda_\theta^{\text{imag}}$ to the corresponding frequency components, and transforms it back to the time domain to obtain a weighted representation $\tilde{x}$, serving as the input for the forecasting model.

### 4.2 MODULE-1: FREQUENCY WEIGHT INITIALIZATION

As discussed in the preliminary analysis, frequency weights have one-one correspondence with kernels. To steer the learning process, we assume the commonly adopted RBF kernel (i.e., $\exp\left(-\gamma\|x_1 - x_2\|^2\right)$, where $\gamma := \frac{1}{2\sigma^2} > 0$ denotes the kernel width) to initialize the non-stationarity measure. Then, the corresponding eigenvalues of RBF kernel naturally serve as the starting point of frequency weights, which can be empirically estimated from the training set. Given $\mathcal{X}_{\text{train}} = \{x^{(i)}\}_{i=1}^N$, we first apply Discrete Fourier Transform (DFT) on each time series sample to obtain the amplitude spectrum $A^{(i)} = \text{Amp}(\text{DFT}(x^{(i)})) \in \mathbb{R}^{L_x \times C}$, where $\text{Amp}(\cdot)$ computes the amplitude. The frequency-wise RBF eigenvalue $\hat{\lambda} \in \mathbb{R}^{L_x \times C}$ is then measured by:

$$\hat{\lambda}(\omega) = \sqrt{\frac{\pi}{\hat{\gamma}}} \exp\left(-\frac{\omega^2}{4\hat{\gamma}}\right), \quad \text{where} \quad \hat{\gamma} = \frac{1}{2\sigma^2(\{A^{(i)}\}_{i=1}^N)}. \tag{1}$$

Here, $\hat{\gamma}$ is the only value to be estimated. Frequency-wise standard deviation $\sigma(\cdot)$ is computed for frequency component $\omega$ over amplitudes of all training samples.

### 4.3 MODULE-2: FREQUENCY WEIGHT OPTIMIZATION

Following the initialization with RBF, one might consider learning the width $\gamma$ for the frequency weights in a data-adaptive way. However, doing so is restricted to the RBF kernel as its functional form remains fixed. By contrast, learning adaptive weights $\lambda_\theta$ corresponds to learning diverse kernel representations itself, which has greater expressiveness than the fixed RBF. Therefore, we learn the frequency weights $\lambda_\theta$ from the RBF initialization by updating the neural networks with stochastic gradient descent. Specifically, we implement Multi-Layer Perceptrons (MLPs) to optimize frequency weights:

$$\lambda_\theta^{\text{real}}(\omega) = \text{ReLU}(\hat{\lambda}(\omega)\mathbf{W}_1^{\text{real}} + \mathbf{b}_1^{\text{real}})\mathbf{W}_2^{\text{real}} + \mathbf{b}_2^{\text{real}},$$
$$\lambda_\theta^{\text{imag}}(\omega) = \text{ReLU}(\hat{\lambda}(\omega)\mathbf{W}_1^{\text{imag}} + \mathbf{b}_1^{\text{imag}})\mathbf{W}_2^{\text{imag}} + \mathbf{b}_2^{\text{imag}}. \tag{2}$$

Here $\mathbf{W}_1^* \in \mathbb{R}^{C \times H}$, $\mathbf{W}_2^* \in \mathbb{R}^{H \times C}$, $\mathbf{b}_1^* \in \mathbb{R}^H$, and $\mathbf{b}_2^* \in \mathbb{R}^C$, where $* \in \{\text{real}, \text{imag}\}$. $H$ is the hidden dimension. Notably, we model the weights for real and imaginary frequency components separately as $\lambda_\theta^{\text{real}}$ and $\lambda_\theta^{\text{imag}} \in \mathbb{R}^{L_x \times C}$. The rationale behind this design is that assigning distinct weights to the real and imaginary components allows the model to capture non-stationarity arising from shifts in both amplitude and phase. In contrast, a single unified weight per frequency can only modulate amplitude, leaving phase-related non-stationarity structures unaddressed.

### 4.4 MODULE-3: FREQUENCY-WEIGHTED REPRESENTATION

Given an input time series $x \in \mathbb{R}^{L_x \times C}$, we first transform it to the frequency domain via DFT, producing real and imaginary coefficients as $F_{\text{real}}, F_{\text{imag}} = \text{DFT}(x)$. Then, the learned frequency weights are applied to the corresponding coefficient using Hadamard product:

$$\tilde{F}_{\text{real}}(\omega) = F_{\text{real}}(\omega) \odot \lambda_\theta^{\text{real}}(\omega), \quad \tilde{F}_{\text{imag}}(\omega) = F_{\text{imag}}(\omega) \odot \lambda_\theta^{\text{imag}}(\omega). \tag{3}$$

The weighted coefficients, i.e., $\tilde{F}_{\text{real}}$ and $\tilde{F}_{\text{imag}}$, are supposed to establish a more stationary representation with enhanced stationary frequencies and suppressed non-stationary ones, accommodating robust forecasting under distribution shifts. By aggregating all weighted frequencies, the final time-domain representation is obtained via Inverse Discrete Fourier Transform (IDFT) as $\tilde{x} = \text{IDFT}(\tilde{F}_{\text{real}} + i\tilde{F}_{\text{imag}}) \in \mathbb{R}^{L_x \times C}$, serving as the input for the downstream forecasting backbone model to perform prediction. The whole framework is trained jointly with the forecasting backbone using mean squared error (MSE) loss in an end-to-end manner.

## 5 EXPERIMENTS

We conduct various experiments on widely adopted benchmark datasets to answer the following questions: **RQ1**: How does FREMEN enhance the performance of existing time series forecasting backbone models? **RQ2**: Does FREMEN mitigate distribution shift issue? **RQ3**: How effective are the learned frequency weights $\lambda_\theta$? **RQ4**: How does each design choice of FREMEN contribute to its performance? **RQ5**: How does the inclusion of FREMEN affect the efficiency of backbone models?

### 5.1 EXPERIMENT SETUP

**Datasets.** We use seven widely adopted datasets in multivariate time series forecasting, including: (1) **ETT** (Electricity Transformer Temperature) with four subsets of oil temperature and electrical load recorded at hourly (ETTh1, ETTh2) and 15-minute (ETTm1, ETTm2) resolutions from July 2016 to July 2018; (2) **ECL** contains 15-minute-level electricity consumption of 321 clients from 2012 to 2014. (3) **Weather** includes 21 meteorological features collected every 10 minutes in 2020. (4) **Traffic** is comprised of hourly-recorded traffic load by 862 sensors in San Francisco freeways from 2015 to 2016. All datasets have been published in (Wu et al., 2021). We adopt the split ratio setting in (Wu et al., 2021), which is 6:2:2 for four ETT datasets and 7:1:2 for the other datasets. A global normalization is applied to transform the whole dataset to a fixed scale. Note that this normalization keeps the statistics unchanged; thus, it is unable to handle non-stationarity.

**Baselines.** We compare FREMEN with state-of-the-art normalization methods for non-stationary time series forecasting including: RevIN (Kim et al., 2021), SAN (Liu et al., 2023), Dish-TS (Fan

Table 2: Forecasting results of backbone models with and without FREMEN. Results are averaged with prediction length $L_y \in \{96, 192, 336, 720\}$. The best results are highlighted in **bold**.

| Methods Metrics | iTransformer MSE | MAE | + FREMEN MSE | MAE | PatchTST MSE | MAE | + FREMEN MSE | MAE | DLinear MSE | MAE | + FREMEN MSE | MAE | RLinear MSE | MAE | + FREMEN MSE | MAE |
|---|---|---|---|---|---|---|---|---|---|---|---|---|---|---|---|---|
| ETTh1 | 0.511 ±0.001 | 0.500 ±0.001 | **0.451** ±0.017 | **0.448** ±0.011 | 0.495 ±0.024 | 0.490 ±0.023 | **0.438** ±0.003 | **0.439** ±0.002 | 0.425 ±0.002 | 0.440 ±0.004 | **0.407** ±0.001 | **0.423** ±0.002 | 0.535 ±0.014 | 0.504 ±0.005 | **0.466** ±0.028 | **0.455** ±0.013 |
| ETTh2 | 0.786 ±0.068 | 0.642 ±0.035 | **0.379** ±0.002 | **0.405** ±0.002 | 0.649 ±0.102 | 0.526 ±0.058 | **0.373** ±0.014 | **0.401** ±0.008 | 0.489 ±0.012 | 0.476 ±0.009 | **0.335** ±0.005 | **0.383** ±0.003 | 0.618 ±0.018 | 0.553 ±0.008 | **0.393** ±0.004 | **0.413** ±0.002 |
| ETTm1 | 0.449 ±0.004 | 0.454 ±0.004 | **0.400** ±0.001 | **0.407** ±0.001 | 0.419 ±0.009 | 0.430 ±0.009 | **0.382** ±0.010 | **0.396** ±0.007 | 0.357 ±0.001 | 0.378 ±0.001 | **0.355** ±0.001 | **0.375** ±0.001 | 0.419 ±0.003 | 0.419 ±0.001 | **0.416** ±0.003 | **0.415** ±0.002 |
| ETTm2 | 0.562 ±0.024 | 0.523 ±0.026 | **0.289** ±0.001 | **0.333** ±0.002 | 0.392 ±0.145 | 0.412 ±0.097 | **0.281** ±0.002 | **0.326** ±0.003 | 0.291 ±0.011 | 0.352 ±0.002 | **0.256** ±0.005 | **0.313** ±0.005 | 0.362 ±0.006 | 0.407 ±0.006 | **0.289** ±0.002 | **0.332** ±0.002 |
| ECL | 0.182 ±0.002 | 0.282 ±0.003 | **0.172** ±0.002 | **0.264** ±0.002 | 0.211 ±0.004 | 0.309 ±0.008 | **0.202** ±0.003 | **0.294** ±0.005 | 0.173 ±0.001 | 0.274 ±0.001 | **0.167** ±0.001 | **0.260** ±0.001 | 0.214 ±0.004 | 0.304 ±0.004 | **0.211** ±0.004 | **0.290** ±0.004 |
| Traffic | 0.571 ±0.007 | 0.314 ±0.007 | **0.429** ±0.013 | **0.285** ±0.009 | 0.594 ±0.006 | **0.315** ±0.006 | **0.512** ±0.009 | 0.329 ±0.009 | 0.453 ±0.000 | 0.318 ±0.000 | **0.436** ±0.001 | **0.298** ±0.002 | 0.629 ±0.008 | 0.390 ±0.003 | **0.622** ±0.010 | **0.376** ±0.007 |
| Weather | 0.252 ±0.003 | 0.300 ±0.007 | **0.251** ±0.003 | **0.276** ±0.002 | **0.248** ±0.004 | 0.301 ±0.007 | 0.256 ±0.005 | **0.279** ±0.003 | 0.245 ±0.001 | 0.298 ±0.002 | **0.226** ±0.001 | **0.265** ±0.001 | 0.269 ±0.001 | 0.319 ±0.001 | **0.258** ±0.003 | **0.283** ±0.004 |

et al., 2023), and FAN (Ye et al., 2024). RevIN, SAN, and Dish-TS conduct time-domain normalization, while FAN focuses on modeling dominant frequencies to overcome non-stationarity.

**Backbones.** FREMEN is a model-agnostic framework that can be applied to any time series forecasting model. To validate its effectiveness, we select four mainstream backbones, including: MLP-based DLinear (Zeng et al., 2023) and RLinear (Li et al., 2023), and Transformer-based iTransformer (Liu et al., 2024) and PatchTST (Nie et al., 2022). The normalization baselines and our FREMEN method are deployed on these backbones for the following experiments.

**Experiments Details.** The prediction length is set as $L_y \in \{96, 192, 336, 720\}$ for all backbones. The input-sequence length $L_x$ is set to 336 for DLinear and 96 for the other backbones. We use the Adam optimizer and report the mean absolute error (MAE) and mean squared error (MSE) as the evaluation metrics. All experiments are implemented with PyTorch 2.3.0 and conducted on a single NVIDIA A100 40GB GPU. Details of setup and full experiment results are in the Appendix.

## 5.2 MAIN RESULTS

**Effectiveness on Time Series Forecasting Backbones.** To answer **RQ1**, we present the multivariate forecasting results in Table 2. Here, the MSE and MAE are presented in the form of mean ± std for five runs across four prediction lengths. It is evident that FREMEN consistently enhances the performance of backbone models by a substantial margin under nearly all experimental settings. For example, the average MSE improvements for iTransformer are notable on ETTh2 (51.78%), ETTm2 (48.57%), and Traffic (24.86%), with an average MSE reduction of 28.43% among all datasets. Comparable improvements are observed for PatchTST, DLinear, and RLinear, with average MSE reductions of 18.75%, 10.31%, and 12.83%, respectively, over all benchmark datasets. The superior results can be primarily attributed to the adaptive frequency weights applied, which yields representations with reduced non-stationarity, thereby benefiting learning of forecasting backbones.

**Comparison with Baseline Methods.** Continuing the investigation of **RQ1**, we present the evaluation result of different normalization methods on iTransformer and DLinear in Table 3. We observe that FREMEN generally outperforms baselines for different forecasting backbones, achieving the best forecasting results in 24 out of 28 experiment settings on average. Specifically, on the ECL dataset, FREMEN achieves MSE values of 0.172 and 0.167 for iTransformer and DLinear, outperforming the best baseline results (i.e., 0.176 by Dish-TS and 0.171 by SAN). Similarly, on the Traffic dataset, the MSE of FREMEN averaged across backbones is 0.432, compared to 0.450, 0.467, 0.491, and 0.456 for RevIN, SAN, FAN, and Dish-TS, respectively. These results may be attributed to the inadequacy of existing methods in modeling non-stationary structures involving spectral shifts.

## 5.3 DETAILED ANALYSIS

**Distribution Shift Analysis.** To answer **RQ2**, we compare the frequency-domain distributional distance between the training and test sets for each normalization method. Specifically, the distance is quantified using the Jensen-Shannon Divergence (JSD) between the empirical distributions of the training and test samples. As shown in Figure 3 (a), we begin by analyzing the JSD across all channels. Overall, FREMEN exhibits the best performance in reducing the train-test distributional gaps.

Table 3: Forecasting results of iTransformer and DLinear with FREMEN and other baseline methods under prediction lengths $L_y \in \{96, 192, 336, 720\}$. The best results are highlighted in **bold**.

| Models | | iTransformer | | | | | | | | | | DLinear | | | | | | | | | |
|---|---|---|---|---|---|---|---|---|---|---|---|---|---|---|---|---|---|---|---|---|---|
| Methods | | FREMEN | | RevIN | | SAN | | FAN | | Dish-TS | | FREMEN | | RevIN | | SAN | | FAN | | Dish-TS | |
| Metrics | | MSE | MAE | MSE | MAE | MSE | MAE | MSE | MAE | MSE | MAE | MSE | MAE | MSE | MAE | MSE | MAE | MSE | MAE | MSE | MAE |
| ETTh1 | 96 | 0.389 | 0.405 | 0.394 | 0.409 | **0.385** | **0.402** | 0.403 | 0.417 | 0.398 | 0.418 | **0.369** | **0.393** | 0.376 | 0.399 | 0.392 | 0.409 | 0.400 | 0.421 | 0.383 | 0.404 |
| | 192 | **0.440** | **0.436** | 0.447 | 0.440 | 0.442 | 0.437 | 0.456 | 0.448 | 0.453 | 0.452 | **0.406** | **0.418** | 0.413 | 0.422 | 0.437 | 0.436 | 0.452 | 0.460 | 0.414 | 0.424 |
| | 336 | 0.479 | 0.459 | 0.490 | 0.463 | **0.473** | **0.453** | 0.512 | 0.485 | 0.499 | 0.481 | **0.425** | **0.431** | 0.430 | 0.432 | 0.450 | 0.446 | 0.489 | 0.488 | 0.441 | 0.444 |
| | 720 | **0.497** | **0.490** | 0.521 | 0.502 | 0.522 | 0.498 | 0.554 | 0.527 | 0.532 | 0.522 | **0.428** | **0.451** | 0.444 | 0.461 | 0.455 | 0.467 | 0.622 | 0.575 | 0.475 | 0.491 |
| ETTh2 | 96 | **0.299** | **0.349** | 0.304 | 0.353 | 0.302 | 0.353 | 0.327 | 0.372 | 0.343 | 0.391 | **0.272** | **0.336** | 0.275 | 0.337 | 0.289 | 0.346 | 0.300 | 0.363 | 0.302 | 0.357 |
| | 192 | **0.376** | **0.396** | 0.391 | 0.405 | 0.382 | 0.399 | 0.425 | 0.438 | 0.533 | 0.506 | **0.329** | **0.375** | 0.335 | 0.376 | 0.356 | 0.385 | 0.408 | 0.439 | 0.418 | 0.423 |
| | 336 | **0.415** | **0.429** | 0.431 | 0.439 | 0.422 | 0.434 | 0.481 | 0.479 | 0.638 | 0.559 | **0.353** | **0.397** | 0.358 | 0.398 | 0.374 | 0.407 | 0.466 | 0.481 | 0.495 | 0.468 |
| | 720 | **0.425** | **0.444** | 0.437 | 0.452 | 0.496 | 0.482 | 0.847 | 0.641 | 0.979 | 0.720 | **0.385** | **0.424** | 0.392 | 0.428 | 0.406 | 0.441 | 0.867 | 0.660 | 0.751 | 0.577 |
| ETTm1 | 96 | **0.329** | **0.367** | 0.339 | 0.374 | 0.336 | 0.375 | 0.342 | 0.377 | 0.340 | 0.378 | **0.293** | **0.341** | 0.301 | 0.343 | 0.295 | 0.343 | 0.319 | 0.359 | 0.300 | 0.344 |
| | 192 | **0.376** | **0.391** | 0.378 | 0.393 | 0.378 | 0.398 | 0.383 | 0.403 | 0.379 | 0.398 | 0.331 | 0.362 | 0.336 | 0.363 | **0.329** | 0.366 | 0.363 | 0.388 | 0.335 | 0.365 |
| | 336 | **0.411** | **0.414** | 0.418 | 0.418 | 0.412 | 0.424 | 0.431 | 0.440 | 0.421 | 0.430 | 0.370 | 0.383 | 0.371 | 0.384 | **0.363** | 0.386 | 0.406 | 0.415 | 0.374 | 0.392 |
| | 720 | 0.483 | 0.455 | 0.489 | **0.454** | 0.478 | 0.459 | 0.491 | 0.477 | 0.492 | 0.470 | 0.426 | 0.415 | 0.428 | 0.417 | **0.414** | 0.417 | 0.473 | 0.458 | 0.436 | 0.434 |
| ETTm2 | 96 | **0.182** | **0.268** | 0.187 | 0.273 | 0.185 | 0.278 | **0.182** | 0.272 | 0.248 | 0.340 | **0.163** | **0.252** | 0.166 | 0.256 | 0.167 | 0.253 | 0.176 | 0.264 | 0.171 | 0.264 |
| | 192 | 0.251 | 0.312 | 0.252 | 0.312 | **0.243** | **0.304** | 0.269 | 0.331 | 0.455 | 0.467 | **0.218** | **0.291** | 0.220 | 0.295 | 0.225 | 0.295 | 0.250 | 0.314 | 0.249 | 0.327 |
| | 336 | **0.311** | **0.348** | 0.314 | 0.351 | 0.326 | 0.362 | 0.383 | 0.406 | 0.449 | 0.447 | **0.273** | **0.326** | 0.276 | 0.327 | 0.282 | 0.331 | 0.323 | 0.363 | 0.324 | 0.370 |
| | 720 | 0.412 | 0.405 | **0.411** | 0.406 | 0.425 | 0.423 | 0.557 | 0.502 | 0.621 | 0.527 | 0.368 | 0.383 | 0.368 | 0.382 | **0.365** | **0.381** | 0.414 | 0.432 | 0.582 | 0.506 |
| ECL | 96 | **0.143** | **0.237** | 0.154 | 0.247 | 0.150 | 0.245 | 0.157 | 0.254 | 0.152 | 0.254 | **0.138** | **0.235** | 0.147 | 0.246 | 0.141 | 0.240 | 0.145 | 0.246 | 0.144 | 0.246 |
| | 192 | **0.161** | **0.252** | 0.167 | 0.257 | 0.164 | 0.258 | 0.169 | 0.267 | 0.164 | 0.264 | **0.153** | **0.247** | 0.160 | 0.258 | 0.157 | 0.255 | 0.161 | 0.262 | 0.160 | 0.261 |
| | 336 | **0.175** | **0.269** | 0.183 | 0.275 | 0.184 | 0.282 | 0.183 | 0.281 | 0.180 | 0.283 | **0.169** | **0.263** | 0.177 | 0.274 | 0.173 | 0.271 | 0.178 | 0.280 | 0.176 | 0.278 |
| | 720 | **0.207** | **0.298** | 0.216 | 0.303 | 0.212 | 0.309 | 0.214 | 0.313 | 0.209 | 0.313 | **0.207** | **0.295** | 0.216 | 0.305 | 0.211 | 0.304 | 0.216 | 0.316 | 0.213 | 0.312 |
| Traffic | 96 | **0.399** | 0.274 | 0.411 | **0.270** | 0.441 | 0.286 | 0.490 | 0.310 | 0.404 | 0.273 | **0.413** | **0.288** | 0.430 | 0.303 | 0.427 | 0.304 | 0.427 | 0.310 | 0.446 | 0.314 |
| | 192 | **0.418** | **0.280** | 0.437 | **0.280** | 0.463 | 0.292 | 0.502 | 0.306 | 0.432 | 0.283 | **0.426** | **0.292** | 0.443 | 0.308 | 0.448 | 0.311 | 0.446 | 0.321 | 0.459 | 0.319 |
| | 336 | **0.432** | **0.285** | 0.450 | 0.286 | 0.480 | 0.298 | 0.524 | 0.319 | 0.453 | 0.291 | **0.439** | **0.298** | 0.455 | 0.314 | 0.468 | 0.319 | 0.462 | 0.329 | 0.472 | 0.326 |
| | 720 | **0.467** | **0.302** | 0.488 | 0.303 | 0.516 | 0.316 | 0.574 | 0.345 | 0.487 | 0.310 | **0.466** | **0.314** | 0.483 | 0.329 | 0.498 | 0.335 | 0.503 | 0.358 | 0.497 | 0.342 |
| Weather | 96 | **0.165** | **0.208** | 0.175 | 0.215 | 0.171 | 0.225 | 0.172 | 0.233 | 0.167 | 0.226 | **0.149** | **0.202** | 0.175 | 0.226 | 0.152 | 0.208 | 0.156 | 0.214 | 0.164 | 0.226 |
| | 192 | **0.215** | **0.252** | 0.225 | 0.257 | 0.219 | 0.269 | 0.232 | 0.291 | **0.215** | 0.268 | **0.193** | **0.242** | 0.217 | 0.260 | 0.199 | 0.256 | 0.204 | 0.265 | 0.202 | 0.262 |
| | 336 | 0.274 | **0.296** | 0.282 | 0.299 | 0.279 | 0.314 | 0.279 | 0.329 | **0.269** | 0.308 | **0.244** | **0.281** | 0.265 | 0.295 | 0.249 | 0.298 | 0.261 | 0.309 | 0.252 | 0.304 |
| | 720 | 0.351 | **0.348** | 0.362 | 0.350 | 0.344 | 0.359 | 0.335 | 0.366 | **0.333** | 0.360 | **0.319** | **0.333** | 0.333 | 0.342 | 0.323 | 0.352 | 0.338 | 0.365 | 0.322 | 0.363 |
| 1st count | | 21 | 23 | 1 | 3 | 4 | 3 | 1 | 0 | 3 | 0 | 24 | 27 | 0 | 1 | 4 | 1 | 0 | 0 | 0 | 0 |

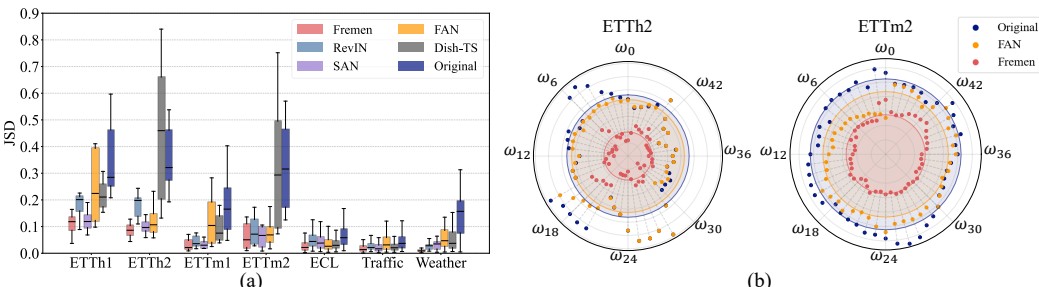

Figure 3: Train-Test distribution distance of each normalization method over (a) channels and (b) frequencies. Each scatter in (b) corresponds to the JSD value of a frequency component. A smaller distance to the center indicates a smaller JSD value.

Furthermore, FREMEN achieves a more compact distance distribution, validating the effectiveness of frequency weights for most channels. We further investigate the improvements in addressing distribution shift from a frequency-domain perspective in Figure 3 (b). The results are evaluated on two datasets with the highest train-test JSD and compared with FAN, which also operates in the frequency domain. The radius of each circle represents the average JSD value on all frequencies. Results suggest that FREMEN significantly reduces distribution shift for almost all frequencies. In contrast, FAN demonstrates noticeable improvements only within a limited frequency range. The superiority of FREMEN stems from its ability to learn stationary representations across the entire frequency spectrum, in contrast to the mechanism of FAN which heuristically models partial spectral.

**Frequency Weight Analysis.** To answer **RQ3**, we compare the learned and the actual stationarity of frequencies on ECL in Figure 4. The blue line represents the averaged amplitude gap of frequencies between the training and test set, which reflects actual extent of non-stationarity. The red line represents the weight learned by FREMEN, which measures the stationarity. For case (a), we observe that the two lines exhibit negative correlation. That is, the weights given by FREMEN is generally higher for frequencies with smaller amplitude gaps and lower for those with larger amplitude gaps. Similar results can be found in case (b), where the weights for the two frequencies with significant amplitude gap are notably lower than those for others. The two examples confirm that FREMEN correctly assigns weights to frequency components. Additionally, we analyze $\lambda_\theta$ via inspecting the corresponding kernel representations. Taking RBF kernels for illustration, Figure 5 shows the kernel shapes corresponding to four selected frequency weights. There is clear evidence that the learned

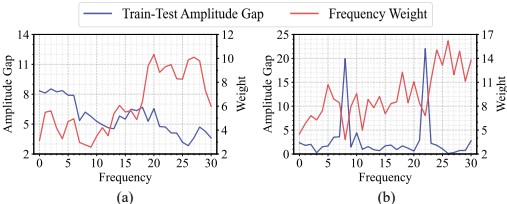

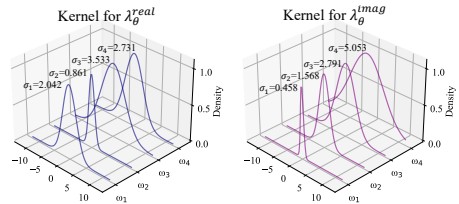

Figure 4: Comparison between the learned frequency weight and the Train-Test amplitude gap.

Figure 5: Kernel representations corresponded to the learned frequency weights.

Figure 6: Ablation study with $L_y \in \{96, 720\}$. Best MSE results are highlighted in **bold**.

| Method | Variant | ETTh1 | | ECL | | Weather | |
|---|---|---|---|---|---|---|---|
| | | 96 | 720 | 96 | 720 | 96 | 720 |
| iTransformer | **+ FREMEN** | **0.389** | **0.497** | **0.143** | **0.207** | **0.165** | **0.351** |
| | *+ random init* | 0.406 | 0.532 | 0.164 | 0.235 | 0.174 | 0.357 |
| | *+ fixed kernel* | 0.416 | 0.548 | 0.166 | 0.237 | 0.177 | 0.359 |
| | *+ single weight* | 0.396 | 0.534 | 0.148 | 0.217 | 0.168 | 0.357 |
| DLinear | **+ FREMEN** | **0.369** | **0.428** | **0.138** | **0.207** | **0.149** | **0.319** |
| | *+ random init* | 0.379 | 0.440 | 0.142 | 0.211 | 0.154 | 0.323 |
| | *+ fixed kernel* | 0.380 | 0.437 | 0.154 | 0.221 | 0.160 | 0.325 |
| | *+ single weight* | 0.376 | 0.433 | 0.141 | 0.209 | 0.152 | 0.321 |

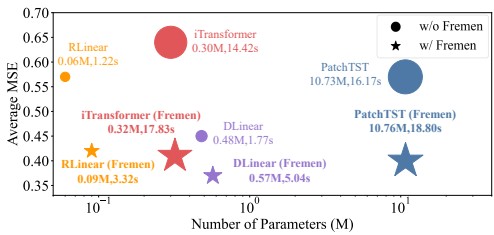

Figure 7: Efficiency analysis of forecasting backbones with (circle) and without FREMEN (star).

kernel representations differ significantly from one another, demonstrating the ability of FREMEN to capture diverse non-stationary patterns in time series data by learning distinct kernels.

**Ablation Study.** To answer **RQ4**, this section systematically evaluates the key components of FREMEN. We consider three variants to assess the contribution of each part: "*random init*" initializes $\hat{\lambda}$ with a random vector; "*fixed kernel*" fixes $\lambda_\theta$ as the RBF eigenvalue; "*single weight*" uses shared weights for real and imaginary frequency coefficients. As shown in Table 6, FREMEN consistently outperforms all variants. Specifically, the increased forecasting error observed in "*random init*" highlights the critical role of initializing with the empirical RBF kernel eigenvalue $\hat{\lambda}$, which provides a meaningful prior and facilitates more effective frequency weight learning. Similar performance degradation are observed for "*single weight*" due to inability in handling phase shifts via separately modeling non-stationarity for real and imaginary frequency components. The most pronounced decline in performance is observed for "*fixed kernel*", which is expected since fixing the kernel form severely restricts the model's expressiveness and adaptability to diverse non-stationarity patterns.

**Model Efficiency Analysis.** To address **RQ5**, we evaluate the efficiency of forecasting backbones integrated with FREMEN, as illustrated in Figure 7 on the ETTh1 and ETTh2 datasets. In this figure, each pattern represents the outcome of a specific experimental setting, with the size of the pattern reflecting the corresponding running time. The results clearly demonstrate that FREMEN is a lightweight yet highly effective enhancement, introducing only a slight increase in the number of parameters (approximately 0.04M) and computational overhead (averaging 2.85 seconds per epoch), while delivering substantial improvements in forecasting performance.

# 6 CONCLUSION

This paper studies the problem of learning robust representations for non-stationary time series forecasting. Existing methods mainly focus on measuring non-stationarity in the time domain using low-order statistics. This paper proposed a novel, non-stationarity-aware representation learning method to capture complex temporal structures and variations. We provided theoretical analysis to show that learning a valid non-stationarity measure in frequencies induced a kernel representation, which can be further represented as an orthonormal set of frequency components weights. We introduced FREMEN, that applied frequency weighting on the input time series to learn a more robust representation for forecasting. FREMEN demonstrated effectiveness via extensive experiments. The results confirmed that FREMEN improved mainstream forecasting models by a large margin and outperformed other state-of-the-art normalization methods.

## 7 REPRODUCIBILITY STATEMENT

We make our code publicly available[2] which contains detailed implementation of our method. The code and hyperparameters for forecasting backbones adopted in this paper are based on the Time-Series-Library[3]. For normalization baseline methods, we utilize the code from their official github repository together with the configurations.

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

## Appendix/Supplementary Materials

## A  Large Language Models (LLMs) Usage Statement

Large Language Models (LLMs) were used solely to assist with the English writing and language polishing of this manuscript. All research ideas, experimental design, data analysis, and scientific content were conceived and executed by the authors without the involvement of LLMs.

# B ADDITIONAL EXPERIMENTAL DETAILS

## B.1 DATASET DETAILS

The statistical properties of the seven benchmark datasets are summarized in Table 4. To quantify the distribution shift between training and test samples, we employ the Jensen-Shannon Divergence (JSD), which serves as a measure of dataset non-stationarity (Mateos et al., 2017). The JSD is computed between the empirical distributions derived from the training and test sets. The computational procedure for JSD between two sample sets is formally described in Algorithm 1. Our implementation utilizes the "jensenshannon" function from the SciPy library, which computes the square root of the JSD. Then, the squared output is considered as the distributional divergence measure.

Table 4: Statistics of benchmark datasets.

| Datasets | ETTh1 | ETTh2 | ETTm1 | ETTm2 | ECL | Traffic | Weather |
|---|---|---|---|---|---|---|---|
| # Channels | 7 | 7 | 7 | 7 | 321 | 862 | 21 |
| # Timesteps | 17,420 | 17,420 | 69,680 | 69,680 | 26,304 | 17,544 | 52,969 |
| Sample Frequency | 1h | 1h | 15mins | 15mins | 1h | 1h | 10mins |
| Time Range | 2016-2017 | 2017-2018 | 2016-2017 | 2017-2018 | 2012-2014 | 2015-2016 | 2020 |
| JSD* | 0.2091 | 0.2839 | 0.1225 | 0.3138 | 0.0716 | 0.0627 | 0.1524 |

\* A smaller JSD indicates a more stationary time series dataset

---

**Algorithm 1:** Computation of Jensen-Shannon Divergence between two sets of samples

**Input**: arrays $a, b$; number of bins $B$
**Output**: Jensen-Shannon Divergence $D_{\mathrm{JSD}}^2$
1: $v_{min} \leftarrow \min(\min(a), \min(b))$
2: $v_{max} \leftarrow \max(\max(a), \max(b))$
3: $h_a \leftarrow \mathrm{histogram}(a; B, [v_{min}, v_{max}])$
4: $h_b \leftarrow \mathrm{histogram}(b; B, [v_{min}, v_{max}])$
5: $p \leftarrow h_a / \sum h_a$
6: $q \leftarrow h_b / \sum h_b$
7: $D_{\mathrm{JSD}} \leftarrow \mathrm{jensenshannon}(p, q, \mathrm{base} = 2)$
8: **return** $D_{\mathrm{JSD}}^2$

---

## B.2 BASELINE METHOD DETAILS

In this study, we study four state-of-the-art normalization methods as baselines: RevIN, SAN, Dish-TS, and FAN. The technical details of each approach are presented below:

**Reversible Instance Normalization (RevIN)** (Kim et al., 2021). RevIN proposes a symmetric normalization-denormalization framework to address distribution shifts in time series data. The method first applies instance-wise z-score normalization to input samples, effectively eliminating non-stationary components. The normalized data is then fed into the forecasting model for forecasting. After which the original statistical properties (i.e., mean and variance) are restored to the forecasting result through a denormalization process. This reversible transformation maintains crucial distributional characteristics while enabling models to operate on stationary representations.

**Slice-level Adaptive Normalization (SAN)** (Liu et al., 2023). SAN introduces a fine-grained normalization approach that operates at the sub-series level instead of the whole input and output series. Unlike direct statistical transfer, SAN employs a dedicated statistics prediction module to explicitly model the evolution of mean and variance of the data distribution. During training, this module is first pre-trained to predict future statistics. Then, it is frozen and used to produce normalized input data for the downstream forecasting models training.

**Dish-TS** (Fan et al., 2023). Dish-TS provides a systematic framework that classifies distribution shifts into intra-space (within input/output spaces) and inter-space (between input/output spaces) variations. The method introduces a specialized network architecture for input and output distribution estimation, augmented with learnable adaptive distribution statistics. Notably, Dish-TS incorporates empirical mean values as prior knowledge to enhance statistical learning.

**Frequency Adaptive Normalization (FAN) (Ye et al., 2024).** FAN represents the first normalization approach that addresses non-stationarity through frequency-domain analysis. The method classifies the whole frequency spectrum into two sets where the top-$k$ dominant frequency components are considered as non-stationary components which are fed into an MLP network to model the future statistical variations. The remaining stationary frequency components are directly fed into the forecasting models for prediction. Combining the two parts of outputs, FAN effectively captures potential variants in frequencies to mitigate non-stationarity.

### B.3 FORECASTING BACKBONE DETAILS

In this work, we evaluate our method on four prominent time series forecasting architectures: the MLP-based DLinear and RLinear, and the Transformer-based iTransformer and PatchTST. We provide an overview of their key design principles:

**DLinear (Zeng et al., 2023).** DLinear establishes a decomposition-based linear architecture. The method first decomposes the input series into trend and seasonal components using moving average smoothing. These components are then processed independently through dedicated linear layers, with their outputs aggregated to produce the final prediction result.

**RLinear (Li et al., 2023).** RLinear adopts an extremely light-weight architecture comprising a single linear layer enhanced with reversible normalization. The approach capitalizes on the inherent capability of linear mappings to capture periodic patterns, while the normalization scheme transforms trend components into seasonality-like representations.

**iTransformer (Liu et al., 2024).** iTransformer reconfigures the standard Transformer architecture for time series analysis. Rather than tokenizing multivariate points at each timestep, it represents entire univariate series as individual tokens. This inverted paradigm enables self-attention mechanisms to focus on cross-variate dependencies while feed-forward networks handle temporal patterns, better accommodating the unique characteristics of time series data.

**PatchTST (Nie et al., 2022).** PatchTST employs a Transformer encoder architecture with two core modifications: First, time series are divided into overlapping or non-overlapping patches that serve as input tokens, reducing sequence length while preserving local patterns. Second, it processes each channel independently with shared weights, enabling efficient multivariate forecasting.

### B.4 OTHER EXPERIMENTAL DETAILS

**Implementation details.** We mainly tune the value of the hidden dimension $H$ within the range of $\{16, 64, 128, 256, 512\}$ and select the one with the best forecasting accuracy on the validation set as the final hyperparameter for each experimental setting. We repeat each experiment for five times with fixed seed and report the average evaluation results.

**Loss Functions.** We employ mean squared error (MSE) as the loss function for all forecasting backbones, which quantifies the averaged squared difference between the predicted and actual target time series. Mathematically, the MSE loss is expressive as: $\mathcal{L}_{\mathrm{MSE}} = \frac{1}{N} \sum_{i=1}^{N} (\hat{y}_i - y_i)^2$, where $N$ is the number of samples, $\hat{y}_i$ represents the predicted value, and $y_i$ represents the actual value.

## C ADDITIONAL EVALUATION RESULTS

### C.1 EFFECTIVENESS ON TIME SERIES FORECASTING BACKBONES

Table 5 presents comprehensive evaluation results across all prediction lengths and benchmark datasets for the four forecasting backbones. The results demonstrate that FREMEN consistently enhances the performance of all four baseline architectures. Quantitative analysis reveals that FREMEN outperforms the original models in 210 out of 224 evaluation scenarios (4 backbones × 7 datasets × 4 prediction lengths × 2 metrics). More specifically, FREMEN achieves average MSE reductions of 12.62%, 12.47%, 17.41%, and 25.97% for prediction lengths of 96, 192, 336, and 720 steps, respectively. Notably, the performance gains become increasingly pronounced as the prediction horizon lengthens. These findings provide strong evidence for the effectiveness of the frequency weighting mechanism in FREMEN, particularly in the context of long-term time series forecasting tasks.

## C.2 COMPARISON WITH BASELINE METHODS

Tables 6 presents comprehensive evaluation results comparing FREMEN against existing normalization approaches on PatchTST and RLinear. The experimental results demonstrate FREMEN's superior performance, particularly on the ETTh2 and ETTm2 datasets which exhibit significant non-stationarity (as quantified by their high JSD values in Table 4). FREMEN achieves remarkable improvements of 17.85% and 19.32% in average MSE reduction on ETTh2 and ETTm2 respectively, significantly outperforming all baseline normalization methods.

Table 5: Full forecasting results of backbones with and without FREMEN under prediction lengths $L_y \in \{96, 192, 336, 720\}$. The best results are highlighted in **bold**.

| Methods Metrics | | iTransformer MSE | MAE | + FREMEN MSE | MAE | PatchTST MSE | MAE | + FREMEN MSE | MAE | DLinear MSE | MAE | + FREMEN MSE | MAE | RLinear MSE | MAE | + FREMEN MSE | MAE |
|---|---|---|---|---|---|---|---|---|---|---|---|---|---|---|---|---|---|
| ETTh1 | 96 | 0.419 | 0.432 | **0.389** | **0.405** | 0.394 | 0.418 | **0.376** | **0.396** | 0.375 | 0.398 | **0.369** | **0.393** | 0.480 | 0.464 | **0.410** | **0.419** |
| | 192 | 0.476 | 0.472 | **0.440** | **0.436** | 0.455 | 0.456 | **0.423** | **0.424** | 0.412 | 0.423 | **0.406** | **0.418** | 0.522 | 0.490 | **0.465** | **0.450** |
| | 336 | 0.547 | 0.525 | **0.479** | **0.459** | 0.512 | 0.503 | **0.468** | **0.451** | 0.438 | 0.444 | **0.425** | **0.431** | 0.556 | 0.510 | **0.497** | **0.466** |
| | 720 | 0.600 | 0.572 | **0.497** | **0.490** | 0.620 | 0.583 | **0.484** | **0.483** | 0.474 | 0.494 | **0.428** | **0.451** | 0.583 | 0.552 | **0.490** | **0.483** |
| ETTh2 | 96 | 0.460 | 0.473 | **0.299** | **0.349** | 0.413 | 0.392 | **0.290** | **0.343** | 0.307 | 0.369 | **0.272** | **0.336** | 0.404 | 0.443 | **0.310** | **0.356** |
| | 192 | 0.656 | 0.593 | **0.376** | **0.396** | 0.479 | 0.473 | **0.367** | **0.392** | 0.402 | 0.431 | **0.329** | **0.375** | 0.530 | 0.511 | **0.397** | **0.408** |
| | 336 | 0.887 | 0.702 | **0.415** | **0.429** | 0.576 | 0.517 | **0.413** | **0.426** | 0.488 | 0.485 | **0.353** | **0.397** | 0.648 | 0.573 | **0.430** | **0.437** |
| | 720 | 1.142 | 0.799 | **0.425** | **0.444** | 1.128 | 0.722 | **0.423** | **0.442** | 0.760 | 0.619 | **0.385** | **0.424** | 0.888 | 0.684 | **0.434** | **0.450** |
| ETTm1 | 96 | 0.382 | 0.411 | **0.329** | **0.367** | 0.373 | 0.401 | **0.317** | **0.359** | 0.300 | 0.343 | **0.293** | **0.341** | 0.364 | 0.386 | **0.354** | **0.382** |
| | 192 | 0.410 | 0.425 | **0.376** | **0.391** | 0.391 | 0.409 | **0.364** | **0.383** | 0.334 | 0.365 | **0.331** | **0.362** | 0.396 | 0.402 | **0.395** | **0.401** |
| | 336 | 0.457 | 0.461 | **0.411** | **0.414** | 0.422 | 0.433 | **0.395** | **0.403** | 0.369 | 0.385 | 0.370 | **0.383** | 0.426 | 0.423 | 0.426 | **0.421** |
| | 720 | 0.546 | 0.518 | **0.483** | **0.455** | 0.491 | 0.478 | **0.453** | **0.440** | 0.424 | 0.420 | 0.426 | **0.415** | **0.488** | 0.463 | 0.490 | **0.454** |
| ETTm2 | 96 | 0.238 | 0.327 | **0.182** | **0.268** | 0.255 | 0.337 | **0.178** | **0.261** | 0.169 | 0.265 | **0.163** | **0.252** | 0.207 | 0.306 | **0.186** | **0.270** |
| | 192 | 0.327 | 0.397 | **0.251** | **0.312** | 0.319 | 0.368 | **0.242** | **0.303** | 0.235 | 0.316 | **0.218** | **0.291** | 0.288 | 0.363 | **0.250** | **0.309** |
| | 336 | 0.617 | 0.581 | **0.311** | **0.348** | 0.463 | 0.456 | **0.305** | **0.344** | 0.304 | 0.365 | **0.273** | **0.326** | 0.392 | 0.435 | **0.310** | **0.346** |
| | 720 | 1.066 | 0.786 | **0.412** | **0.405** | 0.529 | 0.488 | **0.399** | **0.397** | 0.456 | 0.463 | **0.368** | **0.383** | 0.559 | 0.525 | **0.410** | **0.401** |
| ECL | 96 | 0.150 | 0.247 | **0.143** | **0.237** | 0.187 | 0.285 | **0.175** | **0.271** | 0.147 | 0.248 | **0.138** | **0.235** | 0.199 | 0.286 | **0.192** | **0.273** |
| | 192 | 0.163 | 0.262 | **0.161** | **0.252** | 0.193 | 0.293 | **0.186** | **0.280** | 0.160 | 0.261 | **0.153** | **0.247** | 0.198 | 0.289 | **0.193** | **0.275** |
| | 336 | 0.196 | 0.299 | **0.175** | **0.269** | 0.211 | 0.312 | **0.203** | **0.296** | 0.175 | 0.277 | **0.169** | **0.263** | 0.211 | 0.304 | **0.208** | **0.290** |
| | 720 | 0.217 | 0.318 | **0.207** | **0.298** | 0.251 | 0.347 | **0.245** | **0.329** | 0.209 | 0.309 | **0.207** | **0.295** | 0.246 | 0.336 | 0.249 | **0.323** |
| Traffic | 96 | 0.543 | 0.306 | **0.399** | **0.274** | 0.569 | 0.306 | **0.496** | **0.321** | 0.430 | 0.306 | **0.413** | **0.288** | 0.653 | 0.402 | **0.644** | **0.387** |
| | 192 | 0.549 | 0.300 | **0.418** | **0.280** | 0.581 | **0.308** | 0.493 | 0.321 | 0.443 | 0.311 | **0.426** | **0.292** | 0.602 | 0.377 | **0.597** | **0.364** |
| | 336 | 0.583 | 0.316 | **0.432** | **0.285** | 0.595 | **0.314** | 0.511 | 0.327 | 0.456 | 0.319 | **0.439** | **0.298** | 0.609 | 0.379 | **0.605** | **0.365** |
| | 720 | 0.610 | 0.332 | **0.467** | **0.302** | 0.632 | **0.332** | 0.547 | 0.346 | 0.484 | 0.336 | **0.466** | **0.314** | 0.650 | 0.400 | **0.643** | **0.386** |
| Weather | 96 | 0.168 | 0.226 | **0.165** | **0.208** | **0.172** | 0.238 | 0.173 | **0.216** | 0.174 | 0.233 | **0.149** | **0.202** | 0.200 | 0.260 | **0.175** | **0.220** |
| | 192 | 0.218 | 0.280 | **0.215** | **0.252** | **0.209** | 0.268 | 0.218 | **0.255** | 0.216 | 0.275 | **0.193** | **0.242** | 0.240 | 0.297 | **0.223** | **0.262** |
| | 336 | 0.274 | 0.324 | 0.274 | **0.296** | **0.266** | 0.315 | 0.277 | **0.298** | 0.261 | 0.312 | **0.244** | **0.281** | 0.285 | 0.334 | **0.278** | **0.301** |
| | 720 | **0.349** | 0.368 | 0.351 | 0.348 | **0.346** | 0.384 | 0.355 | **0.348** | 0.328 | 0.370 | **0.319** | **0.333** | 0.349 | 0.384 | 0.356 | **0.350** |

Table 6: Forecasting results of PatchTST and RLinear with FREMEN and other baseline methods under prediction lengths $L_y \in \{96, 192, 336, 720\}$. The best results are highlighted in **bold**.

| Models | | PatchTST | | | | | | | | | | RLinear | | | | | | | | | |
|---|---|---|---|---|---|---|---|---|---|---|---|---|---|---|---|---|---|---|---|---|---|
| Methods Metrics | | FREMEN MSE | MAE | RevIN MSE | MAE | SAN MSE | MAE | FAN MSE | MAE | Dish-TS MSE | MAE | FREMEN MSE | MAE | RevIN MSE | MAE | SAN MSE | MAE | FAN MSE | MAE | Dish-TS MSE | MAE |
| ETTh1 | 96 | **0.376** | **0.396** | 0.379 | 0.399 | 0.385 | 0.400 | 0.394 | 0.409 | 0.400 | 0.419 | **0.410** | **0.419** | 0.435 | 0.435 | 0.418 | 0.426 | 0.411 | 0.423 | 0.482 | 0.476 |
| | 192 | **0.423** | **0.424** | 0.426 | 0.432 | 0.438 | 0.435 | 0.452 | 0.445 | 0.452 | 0.453 | **0.465** | **0.450** | 0.479 | 0.458 | 0.467 | 0.453 | 0.467 | 0.457 | 0.535 | 0.507 |
| | 336 | **0.468** | **0.451** | 0.469 | 0.457 | 0.490 | 0.462 | 0.508 | 0.482 | 0.543 | 0.511 | **0.497** | **0.466** | 0.517 | 0.476 | 0.506 | 0.472 | 0.525 | 0.496 | 0.579 | 0.530 |
| | 720 | **0.484** | **0.483** | 0.528 | 0.510 | 0.548 | 0.513 | 0.558 | 0.528 | 0.672 | 0.600 | **0.490** | **0.483** | 0.517 | 0.497 | 0.502 | 0.488 | 0.568 | 0.537 | 0.612 | 0.571 |
| ETTh2 | 96 | **0.290** | **0.343** | 0.292 | 0.345 | 0.309 | 0.359 | 0.335 | 0.377 | 0.340 | 0.383 | **0.310** | **0.356** | 0.322 | 0.366 | 0.313 | 0.361 | 0.337 | 0.379 | 0.444 | 0.468 |
| | 192 | **0.367** | **0.392** | 0.379 | 0.403 | 0.413 | 0.420 | 0.420 | 0.436 | 0.393 | 0.393 | 0.397 | **0.408** | 0.406 | 0.415 | **0.393** | 0.412 | 0.428 | 0.440 | 0.588 | 0.542 |
| | 336 | **0.413** | **0.426** | 0.425 | 0.440 | 0.490 | 0.474 | 0.472 | 0.477 | 0.558 | 0.507 | **0.430** | **0.437** | 0.443 | 0.445 | 0.436 | 0.445 | 0.484 | 0.477 | 0.742 | 0.614 |
| | 720 | **0.423** | **0.442** | 0.441 | 0.458 | 0.491 | 0.483 | 0.796 | 0.627 | 0.821 | 0.605 | **0.434** | **0.450** | 0.444 | 0.456 | 0.442 | 0.460 | 0.833 | 0.637 | 1.202 | 0.783 |
| ETTm1 | 96 | **0.317** | **0.359** | 0.322 | 0.361 | 0.329 | 0.372 | 0.339 | 0.375 | 0.351 | 0.391 | **0.354** | **0.382** | 0.366 | 0.384 | 0.368 | 0.393 | 0.365 | 0.393 | 0.365 | 0.395 |
| | 192 | **0.364** | **0.383** | 0.366 | 0.385 | 0.366 | 0.387 | 0.381 | 0.399 | 0.383 | 0.409 | 0.395 | **0.401** | 0.402 | 0.401 | **0.392** | 0.404 | 0.410 | 0.419 | 0.400 | 0.412 |
| | 336 | **0.395** | **0.403** | 0.397 | 0.408 | 0.396 | 0.407 | 0.426 | 0.433 | 0.410 | 0.430 | **0.426** | **0.421** | 0.434 | 0.422 | 0.426 | 0.426 | 0.452 | 0.446 | 0.432 | 0.436 |
| | 720 | **0.453** | **0.440** | 0.455 | 0.444 | 0.456 | 0.442 | 0.495 | 0.476 | 0.479 | 0.473 | **0.490** | **0.454** | 0.495 | 0.455 | 0.491 | 0.460 | 0.519 | 0.491 | 0.501 | 0.476 |
| ETTm2 | 96 | **0.178** | **0.261** | 0.181 | 0.267 | **0.178** | 0.263 | 0.187 | 0.279 | 0.203 | 0.300 | 0.186 | **0.270** | 0.189 | 0.273 | **0.184** | 0.272 | 0.188 | 0.278 | 0.217 | 0.316 |
| | 192 | **0.242** | **0.303** | 0.245 | 0.305 | 0.255 | 0.320 | 0.265 | 0.332 | 0.330 | 0.390 | **0.250** | **0.309** | 0.253 | 0.312 | **0.250** | 0.311 | 0.284 | 0.344 | 0.311 | 0.379 |
| | 336 | **0.305** | **0.344** | 0.309 | 0.348 | 0.359 | 0.371 | 0.407 | 0.418 | 0.453 | 0.461 | 0.310 | **0.346** | 0.314 | 0.349 | **0.309** | 0.347 | 0.397 | 0.407 | 0.490 | 0.479 |
| | 720 | **0.399** | **0.397** | 0.417 | 0.412 | 0.410 | 0.410 | 0.570 | 0.516 | 0.650 | 0.564 | 0.410 | **0.401** | 0.414 | 0.404 | **0.407** | 0.406 | 0.573 | 0.512 | 1.096 | 0.689 |
| ECL | 96 | **0.175** | **0.271** | 0.180 | 0.279 | 0.185 | 0.277 | 0.181 | **0.271** | 0.186 | 0.289 | 0.192 | **0.273** | 0.201 | 0.280 | 0.190 | 0.278 | **0.187** | 0.276 | 0.192 | 0.284 |
| | 192 | **0.186** | **0.280** | 0.188 | 0.285 | 0.190 | 0.282 | 0.187 | 0.277 | 0.192 | 0.294 | **0.193** | **0.275** | 0.200 | 0.283 | **0.193** | 0.281 | 0.194 | 0.285 | 0.197 | 0.289 |
| | 336 | 0.203 | 0.296 | 0.209 | 0.306 | 0.205 | 0.298 | **0.202** | **0.294** | 0.207 | 0.311 | 0.208 | **0.290** | 0.215 | 0.298 | 0.207 | 0.297 | **0.205** | 0.298 | 0.210 | 0.304 |
| | 720 | 0.245 | 0.329 | 0.245 | 0.330 | 0.244 | 0.329 | **0.239** | **0.327** | 0.243 | 0.340 | 0.249 | **0.323** | 0.256 | 0.330 | 0.247 | 0.330 | **0.243** | 0.332 | 0.245 | 0.336 |
| Traffic | 96 | 0.496 | 0.321 | 0.485 | **0.297** | 0.515 | 0.322 | 0.535 | 0.325 | **0.471** | 0.300 | 0.644 | **0.387** | 0.648 | 0.388 | **0.620** | 0.394 | 0.670 | 0.423 | 0.650 | 0.406 |
| | 192 | 0.493 | 0.321 | 0.496 | **0.304** | 0.523 | 0.324 | 0.543 | 0.323 | **0.483** | 0.304 | 0.597 | 0.364 | 0.601 | 0.365 | 0.595 | 0.373 | **0.566** | 0.370 | 0.606 | 0.382 |
| | 336 | 0.511 | 0.327 | 0.519 | **0.307** | 0.540 | 0.330 | 0.558 | 0.329 | **0.493** | 0.309 | 0.605 | **0.365** | 0.608 | 0.368 | 0.602 | 0.375 | **0.580** | 0.374 | 0.613 | 0.384 |
| | 720 | 0.547 | 0.346 | 0.559 | **0.325** | 0.582 | 0.350 | 0.608 | 0.348 | **0.530** | 0.329 | 0.643 | **0.386** | 0.647 | 0.394 | 0.658 | 0.394 | **0.626** | 0.394 | 0.655 | 0.404 |
| Weather | 96 | 0.173 | 0.216 | **0.166** | **0.212** | 0.172 | 0.225 | 0.170 | 0.230 | 0.167 | 0.232 | **0.175** | **0.220** | 0.198 | 0.238 | 0.181 | 0.230 | 0.188 | 0.240 | 0.193 | 0.267 |
| | 192 | 0.218 | 0.255 | **0.213** | **0.252** | 0.215 | 0.268 | 0.215 | 0.277 | 0.214 | 0.282 | **0.223** | **0.262** | 0.242 | 0.272 | 0.227 | 0.270 | 0.229 | 0.279 | 0.242 | 0.312 |
| | 336 | 0.277 | 0.298 | 0.277 | **0.297** | 0.268 | 0.303 | **0.265** | 0.313 | 0.283 | 0.342 | 0.278 | **0.301** | 0.292 | 0.307 | 0.281 | 0.312 | **0.276** | 0.319 | 0.290 | 0.349 |
| | 720 | 0.355 | 0.348 | 0.353 | **0.348** | 0.339 | 0.362 | **0.334** | 0.355 | 0.351 | 0.388 | 0.356 | **0.350** | 0.364 | 0.353 | 0.358 | 0.365 | **0.343** | 0.368 | 0.360 | 0.400 |
| 1st count | | 18 | **19** | 2 | 8 | 1 | 0 | 4 | 3 | 4 | 1 | 14 | 28 | 0 | 1 | 9 | 0 | 8 | 0 | 0 | 0 |

## C.3 FREQUENCY-WISE DISTRIBUTION SHIFT ANALYSIS

Figure 8 provides additional examples illustrating the frequency-wise train-test distribution discrepancies. Although variables across different datasets exhibit diverse spectral distributions, FREMEN demonstrates effectiveness in addressing distributional differents between the training and test set.

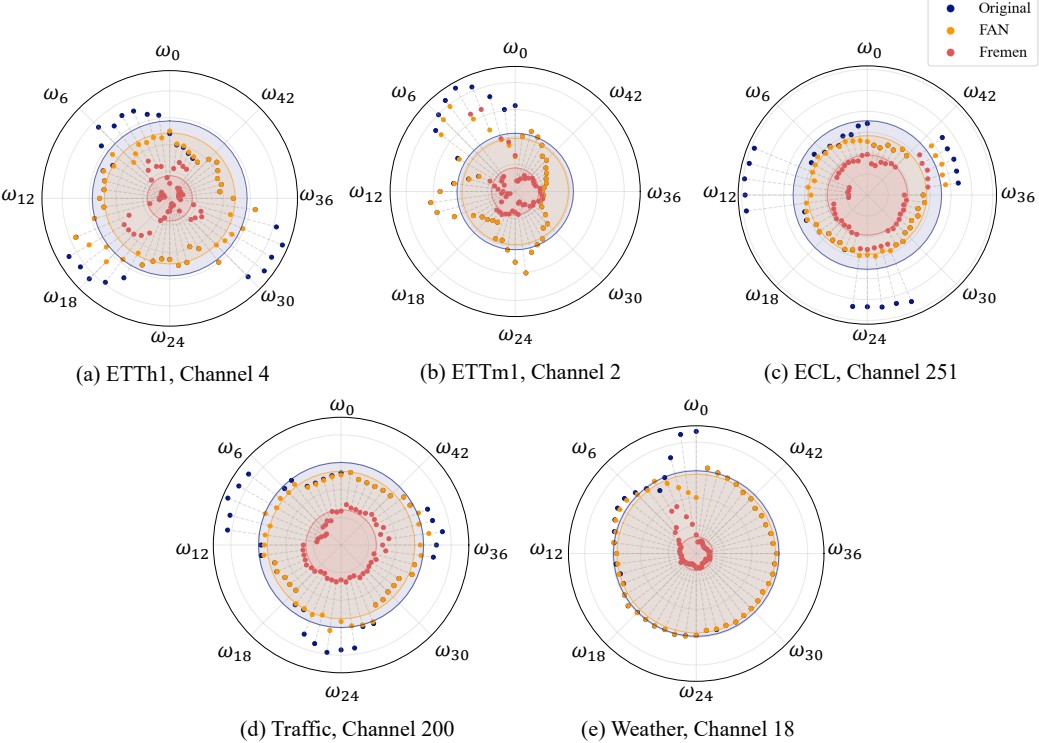

(a) ETTh1, Channel 4          (b) ETTm1, Channel 2          (c) ECL, Channel 251

(d) Traffic, Channel 200          (e) Weather, Channel 18

Figure 8: Additional showcases on the Train-Test distribution distance over all frequencies. Each scatter corresponds to the JSD value of a frequency component. A smaller distance to the center indicates a smaller JSD value.

## C.4 FREQUENCY WEIGHT ANALYSIS

To investigate the dynamics of the learned frequency weights, we visualize both the training loss and the evolution of $\lambda_\theta^{\text{real}}$ in Figure 9, using DLinear on the Traffic dataset. The upper subfigure demonstrates a clear positive correlation between increasing $\lambda_\theta^{\text{real}}$ and improved model accuracy. As discussed in our preliminary analysis from the main paper, $\lambda_\theta^{\text{real}}$ governs kernel representation in the frequency domain. While its exact closed-form kernel mapping remains analytically intractable, we hypothesize its membership within common kernel families and empirically analyze the induced representational transformations. The lower subfigure illustrates the kernel evolution under two constrained settings: RBF and Cauchy kernels, sampled across five training epochs. Notably, we observe divergent trends in the scale parameters $\sigma$: the RBF kernel's $\sigma$ exhibits monotonic growth, while the Cauchy kernel's $\sigma$ demonstrates consistent decay. This antithetical behavior underscores the expressive flexibility of $\lambda_\theta^{\text{real}}$, validating its capacity to adaptively model heterogeneous non-stationary patterns through implicit kernel learning.

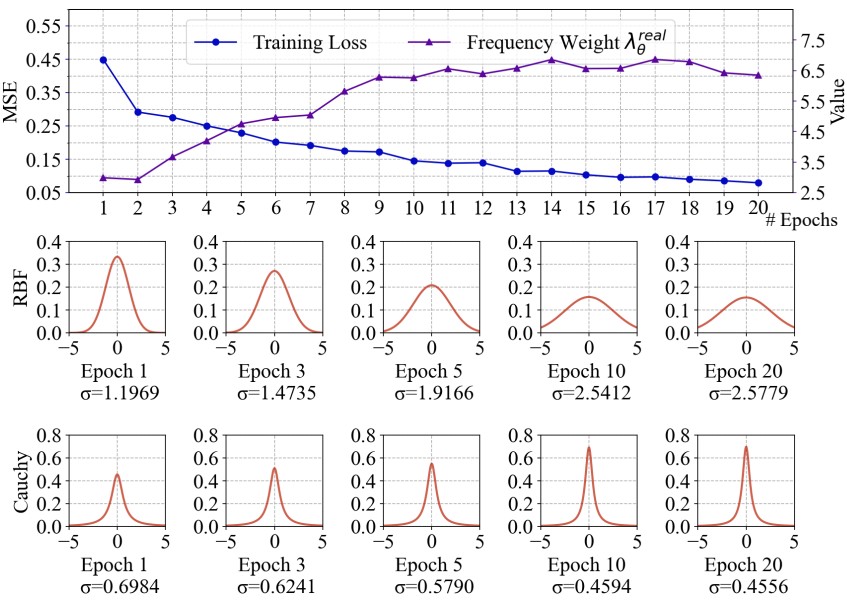

Figure 9: Training process of DLinear with FREMEN on the Traffic dataset. Upper subgraph: the change of the training loss and weights for frequency components. Lower subgraph: the evolution of kernels corresponding to the frequency weights.

