# OpenReview forum: "Measuring Frequency Non-Stationarity for Robust Time Series Forecasting"
_ICLR.cc/2026/Conference — ICLR 2026 Conference Withdrawn Submission_

### Official Review · Reviewer_CDmr · 2025-10-29

**Soundness:** 2
**Presentation:** 3
**Contribution:** 2
**Rating:** 4
**Confidence:** 4

**Summary:**

The paper addresses the challenge of distribution shifts between training and test sets in non-stationary time series forecasting. Unlike conventional normalization methods that remove low-order time-domain statistics—often insufficient to capture deeper non-stationary structures—the authors propose a spectral analysis approach to assess the stationarity of each frequency component across distributions. By adaptively down-weighting unstable frequencies, the method effectively reduces distributional discrepancies and improves model robustness. Experimental results further demonstrate its solid performance.

**Strengths:**

1. The paper clearly identifies the issue of distribution shift between training and test sets, which is an important and practical challenge in time series forecasting.
2. It introduces an adaptive reweighting mechanism to mitigate this problem, achieving consistently superior experimental results.
3. The mathematical formulation is well detailed and presented with clear derivations.

**Weaknesses:**

1. The authors argue that low-order statistics are insufficient to characterize complex distributional structures and temporal dependencies (L72–L74). However, many prior works using such statistics still achieve competitive or even superior results (e.g. SAN, FAN, DDN [1])compared to those reported here. What accounts for this performance gap and could this paper's idea can be used to improve that low-order statistics normalization methods?
2. The proposed method appears to be largely built upon FAN. I wonder whether the implementation also inherits FAN's known issue of potential test-set information leakage.
3. In Table 2, DLinear uses a prediction length of 336, whereas PatchTST remains at 96. For fair comparison—especially when positioning this work as a plug-in—maintaining the original input length would be preferable.
4. This work seems to blend elements of FAN and [2] Could the authors clarify whether there are essential design differences? While the motivation is valid, I remain concerned that distribution shifts between training and test sets are inherent and unavoidable. Excessive down-weighting of non-stationary components might remove meaningful temporal variations, as non-stationarity is a natural and persistent property rather than pure random.

[1] DDN: Dual-domain dynamic normalization for non-stationary time series forecasting
[2] Frequency-domain MLPs are More Effective Learners in Time Series Forecasting.

**Questions:**

see weeknesses

---

### Official Review · Reviewer_isEN · 2025-10-30

**Soundness:** 2
**Presentation:** 3
**Contribution:** 2
**Rating:** 4
**Confidence:** 4

**Summary:**

This paper argues that conventional normalization methods for non-stationary time series forecasting typically rely on low-order time-domain statistics, which may be insufficient to capture the intrinsic non-stationary structures of the data. To address this limitation, the authors propose FREMEN, a frequency-domain reweighting method that measures the degree of non-stationarity for each frequency component and reweights them to form a more stationary representation. Compared with mainstream normalization techniques and forecasting backbones, FREMEN achieves notable performance improvements across multiple datasets.

**Strengths:**

- The core idea of modeling and correcting non-stationarity in the frequency domai by assigning adaptive weights to individual frequency components is intuitive and easy to follow.
- The proposed FREMEN approach achieves competitive and consistent performance across various benchmarks. The authors provide adequate visualization analyses illustrating how the method mitigates distribution shift, as well as ablation studies that validate the importance of its key components.

**Weaknesses:**

- The connection between metioned Yaglom’s Theorem and the proposed FREMEN framework remains vague. The paper would benefit from a more detailed and accessible explanation of how Yaglom’s Theorem theoretically supports the frequency-wise modeling of non-stationarity, and how it justifies the design of the reweighting mechanism.
- The visualization analyses suggest that FREMEN effectively reduces the input series’ distribution shift between training and testing data. However, non-stationarity also exists in the target series, which plays a crucial role in forecasting accuracy. It is unclear how (or whether) FREMEN addresses this aspect of non-stationarity. Additional discussion or experiments on this point would strengthen the contribution.
- As shown in Table 6, FREMEN with randomly initialized $\lambda$ exhibits a noticeable performance drop. It is recommended to provide deeper theoretical or empirical analysis to explain this sensitivity.

**Questions:**

Please refer to the weaknesses part.

---

### Official Review · Reviewer_BrRQ · 2025-10-30

**Soundness:** 3
**Presentation:** 3
**Contribution:** 2
**Rating:** 4
**Confidence:** 4

**Summary:**

This paper addresses the challenge of distribution shift between training and test data caused by non-stationarity in time series. Unlike traditional approaches that rely heavily on low-order statistics and struggle to capture distributional changes, the paper employs spectral analysis to measure non-stationarity. The resulting statistics are used to reweight different non-stationary components, thereby reducing the distribution discrepancy between training and test sets. Experiments demonstrate strong performance, validating the effectiveness of the method.

**Strengths:**

1. Clear and reasonable motivation, easy to understand.
2. The method is intuitive, simple, and efficient to implement.
3. Strong experimental performance, especially on the Traffic dataset.

**Weaknesses:**

1. The paper should more comprehensively discuss recent works on non-stationarity handling to better position its contributions.
2. The method appears to be a relatively simple reweighting extension over FAN, combined with a distribution prediction module similar to prior normalization-based approaches (e.g. SAN), which may weaken novelty.
3. The paper lacks ablation studies for individual components, which limits the clarity of the source of gains.
4. Given the inherent presence of distribution shift, if no data leakage exists, how does the method ensure the frequency weighting learned from training data remains aligned with true future non-stationary components? To what extent can higher-order spectral statistics provide consistent weighting? Some theoretical or empirical analysis would help.

**Questions:**

1. When applying spectral analysis for non-stationarity detection, how does the method avoid data leakage?
2. See weaknesses.

**Details Of Ethics Concerns:**

nan

---

### Official Review · Reviewer_BVCB · 2025-11-01

**Soundness:** 1
**Presentation:** 2
**Contribution:** 3
**Rating:** 4
**Confidence:** 3

**Summary:**

This paper addresses the challenge of non-stationarity in time series forecasting, which often causes poor generalization due to distribution shifts between training and test data. The authors propose FREMEN, a frequency-space, non-stationarity-aware normalization method that measures stationarity in the frequency domain. By downweighting non-stationary frequency components through a kernel representation induced by the Fourier transform, FREMEN effectively mitigates distributional discrepancies. Experiments across multiple benchmark datasets and forecasting models demonstrate significant improvements in predictive performance and robustness.

**Strengths:**

1. Using spectral analysis to measure stationarity and to learn frequency weights is innovative.
2. Extensive experiments demonstrate the algorithm’s effectiveness in modeling non-stationarity.

**Weaknesses:**

1. The paper lacks rigor. It is unclear how Equation (1) is derived, as no corresponding mathematical derivation is provided.
2. The authors use RevIN for normalization in the code, but it is not explained in the main text. Moreover, directly comparing with the backbone without normalization is unfair.

**Questions:**

see weaknesses

---

### Note · Authors · 2025-11-27

I have read and agree with the venue's withdrawal policy on behalf of myself and my co-authors.